



# A Satellite Data-Driven Framework to Rapidly Quantify Air Basin-Scale NO$_x$ Emission and Its Application to the Po Valley during the COVID-19 Pandemic

Kang Sun[1,2], Lingbo Li[1], Shruti Jagini[1], and Dan Li[3]

[1]Department of Civil, Structural and Environmental Engineering, University at Buffalo, Buffalo, NY, USA
[2]Research and Education in Energy, Environment and Water Institute, University at Buffalo, Buffalo, NY, USA
[3]Department of Earth and Environment, Boston University, Boston, MA, USA

**Correspondence:** Kang Sun (kangsun@buffalo.edu)

**Abstract.** The evolving nature of the COVID-19 pandemic necessitates timely estimates of the resultant perturbations to anthropogenic emissions. Here we present a novel framework based on the relationships between observed column abundance and wind speed to rapidly estimate air basin-scale NO$_x$ emission rate and apply it at the Po Valley in Italy using OMI and TROPOMI NO$_2$ tropospheric column observations. The NO$_x$ chemical lifetime is retrieved together with the emission rate and found to be 15–20 h in winter and 5–6 h in summer. A statistical model is trained using the estimated emission rates before the pandemic to predict the trajectory without COVID-19. Compared with this business-as-usual trajectory, the real 2020 emission rates show two distinctive drops in March ($-41\%$) and November ($-35\%$) that correspond to tightened COVID-19 control measures. The temporal variation of pandemic-induced NO$_x$ emission changes qualitatively agree with Google and Apple mobility indicators. The overall net NO$_x$ emission reduction in 2020 due to the COVID-19 pandemic is estimated to be $21\%$.

## 1 Introduction

Satellite observations have revolutionized our ability to observe the Earth's atmospheric composition and air quality. Vertical column densities (VCDs) of reactive species such as NO$_2$, HCHO, SO$_2$, and NH$_3$ are retrieved from the observed radiances in ultraviolet, visible, or the infrared bands. The tropospheric VCD (TVCD) retrieval of NO$_2$ has been widely used to infer the emissions of nitrogen oxides (NO$_x$ = NO$_2$ + NO), which is at the center stage of atmospheric chemistry by modulating ozone and secondary aerosol formation (Kroll et al., 2020). The NO$_x$ emissions are dominated by anthropogenic fossil fuel combustion, and its chemical lifetime in the lower troposphere is relatively short. Consequently, the satellite-observed NO$_2$ TVCD is highly responsive to perturbations of human activities, including economic recession (Castellanos and Boersma, 2012; Russell et al., 2012), long- and short-term emission regulations (Duncan et al., 2016; Mijling et al., 2009; Witte et al.,



2009), and the ongoing global pandemic caused by coronavirus, COVID-19 (Bauwens et al., 2020; Liu et al., 2020; Huang and Sun, 2020).

Although $NO_2$ TVCD is well established as an indicator of $NO_x$ emission, the quantitative connection between $NO_2$ abundance and $NO_x$ emission is confounded by nonlinear chemistry and meteorology (Valin et al., 2014; Goldberg et al., 2020; Keller et al., 2020). Many $NO_x$ emission inference methods have been proposed using chemical transport models (CTMs)
that resolve chemistry and meteorology in space and time, including mass balance (Martin et al., 2003; Lamsal et al., 2011; Zheng et al., 2020), four-dimension variational data assimilation (4D-Var, Qu et al., 2019; Wang et al., 2020), and Kalman filters (Miyazaki et al., 2020a; Mijling and Van Der A, 2012; Ding et al., 2020). Ding et al. (2020) and Miyazaki et al. (2020b) used CTMs to estimate $NO_x$ emission reduction in China in the early phase of the COVID-19 pandemic, but it is a growing challenge to match the resolution, lag time, and running cost of CTMs with the new generation satellite products that resolve
the $NO_2$ spatial distribution down to a few km. As such, observational data-driven approaches have been also developed, which attempt to derive emissions based on the observed column abundance and without invoking CTMs. A common way to estimate emissions of short-lived species like $NO_x$ is to retrieve emission and lifetime simultaneously by fitting an exponentially-modified Gaussian (EMG) function to the downwind plumes from relatively isolated emission sources (e.g., cities or power plants) (Beirle et al., 2011; Liu et al., 2016; de Foy et al., 2015; Lu et al., 2015; Goldberg et al., 2019b, a; Laughner and Cohen,
2019; Valin et al., 2013; Zhang et al., 2019). However, the observational data-driven approaches using OMI only provide warm-season or annually averaged emissions and hence cannot capture the rapidly varying and ongoing COVID-19-induced emission changes. The availability of much more finely resolved TROPOMI observations since 2018 enables observation-based $NO_x$ emission estimates at daily scale over a megacity (Lorente et al., 2019).

Based on satellite observations and reanalysis wind speed, we develop a novel framework that directly and quickly quantifies
air basin-scale $NO_x$ emissions at monthly resolution. We demonstrate this framework using $NO_2$ TVCDs from both OMI and TROPOMI over the Po Valley air basin in Italy, which has been severely affected by COVID-19 (Filippini et al., 2020). The COVID-19-induced emission decline has to be disentangled from pre-existing trends and seasonality. Leveraging the long data record from OMI, we build a statistical model using historic emission rates and predict the business-as-usual trajectory in 2020–2021. The difference between this trajectory and the real 2020–2021 emissions reflects the net effect of COVID-19. As
the pandemic and the controlling policies are still evolving in 2021, this extrapolation using long-term satellite record offers significant advantage over a simple 2020 vs. 2019 comparison. Although only $NO_x$ emission in the Po Valley is investigated in this work, this satellite data-driven framework can be readily applied to other satellite products and regions to rapidly characterize emission changes.

## 2 Materials

### 2.1 Satellite TVCDs

We use the most recent (version 4) $NO_2$ level 2 TVCD retrievals from the NASA operational standard product for OMI (Lamsal et al., 2020). The operational TROPOMI $NO_2$ product (van Geffen et al., 2020; ESA, 2018) used in this study underwent





several algorithm updates since its public release in 30 April 2018. A significant cloud retrieval algorithm update happened in November 2020, leading to substantial increase of retrieved $NO_2$ TVCD in polluted regions. The TROPOMI $NO_2$ algorithm is

expected to be updated with full reprocessing in 2021 to improve its consistency and continuity (GES DISC, 2021). The level 2 orbits covering the geographical region of interest over every month are standardized into single files from October 2004 to February 2021 for OMI and from May 2018 to February 2021 for TROPOMI. We only use quality-assured level 2 pixels with cloud fraction $< 0.3$ and solar zenith angle $< 70°$. Throughout the OMI mission, its across-track pixels are limited to 5–23 out of 1–60 to avoid the row anomaly and keep the time series analysis consistent (Duncan et al., 2016). TROPOMI features

450 pixels across its 2600-km swath and a nadir pixel size of $3.5 \times 5.5$ km$^2$ ($3.5 \times 7$ km$^2$ before 6 August 2019), leading to significantly higher spatial resolution than OMI, whose nadir pixel size is $13 \times 24$ km$^2$.

Validation studies of both OMI and TROPOMI $NO_2$ TVCDs consistently show systematic low biases (Choi et al., 2020; Judd et al., 2020; Verhoelst et al., 2021), which can be largely attributed to the horizontally coarse a priori profile representation in the air mass factor (AMF) calculation. This low bias matters less for emission trend analysis, but will proportionally impact

the absolute values of the derived emission rate. This study focuses on an air basin in which high level of pollution is confined, and the spatial gradient is significantly less than many other polluted regions. The relative biases between OMI and TROPOMI $NO_2$ TVCD are assessed by comparing strictly collocated level 2 retrievals and given in Appendix A. OMI $NO_2$ TVCD is generally higher than TROPOMI in the cold season with monthly OMI-TROPOMI normalized mean bias (NMB) up to over 30%, whereas the TROPOMI TVCD is generally higher in the warm season with monthly OMI-TROPOMI NMB down to

$-20\%$.

## 2.2   Study domain and $NO_x$ emission inventories

The Po Valley air basin is delineated according to the boundary between the flat terrain in northern Italy and mountain ranges in the north, west, and south as well as the Adriatic Sea coastline in the east, as shown in Figure 1. The air basin area is $6.6 \times 10^4$ km$^2$. The west-east length scale is $\sim 500$ km, and the south-north length scale is $\sim 300$ km, both larger than the square root of

basin area (257 km) due to the irregularity of the basin shape. We contrast our derived monthly, air basin-scale $NO_x$ emission rates with four global inventories. Their emission distributions near the Po Valley air basin are illustrated in Figure 1. The Jet Propulsion Laboratory (JPL) chemical reanalysis provides monthly top-down emission estimates at $1.1° \times 1.1°$ spatial resolution in 2005–2019 (Miyazaki et al., 2019, 2020a). The $NO_x$ emissions from the JPL chemical reanalysis (Figure 1a) are constrained by assimilating $O_3$, $NO_2$, CO, $HNO_3$, and $SO_2$ from the OMI, GOME-2, SCIAMACHY, MLS, TES, and

MOPITT satellite instruments (Miyazaki et al., 2020a) and are considered to have the highest accuracy in spite of its relatively low spatial resolution. The rest three are bottom-up emission inventories, including the Community Emission Data System (CEDS, McDuffie et al., 2020), Emissions Database for Global Atmospheric Research version 4.3.2 (EDGAR, Crippa et al., 2018), and the Peking University $NO_x$ (PKUNOx, Huang et al., 2017) inventories. The CEDS inventory is spatially resolved at $0.5° \times 0.5°$ (Figure 1b) and available monthly from 1970 to 2017. Both EDGAR and PKUNOx are at $0.1° \times 0.1°$ spatial

resolution (Figure 1c-d). EDGAR is available annually from 1970 to 2012, and PKUNOx is available monthly from 1960 to



2014. Because of the large the grid sizes of JPL chemical reanalysis and CEDS inventories, we calculate the air basin-mean emission rate by averaging inventory grid cells that overlap with the Po Valley air basin, weighted by the overlapping area.

**Figure 1.** Spatial distribution of annual $NO_x$ emission in 2005 near the Po Valley air basin (black dashed line) from (a) JPL chemical reanalysis, (b) CEDS, (c) EDGAR, and (d) PKUNOx.

## 2.3 Wind fields

We use wind fields gridded at $0.25° \times 0.25°$ spatial resolution and hourly temporal resolution from the ERA5 reanalysis meteorology (Hersbach et al., 2020). The relevant ERA5 fields are spatiotemporally interpolated at each individual OMI/TROPOMI level 2 observation. Previous observational data-driven emission inference studies represented horizontal advection of $NO_2$ (or similar short-lived tracers like $SO_2$ and $NH_3$) by 10m wind above the surface (de Foy et al., 2015), 100m above the surface (Goldberg et al., 2020), vertically averaged wind from surface to 500m (Lu et al., 2015; Liu et al., 2016; Goldberg et al., 2019a), or vertically averaged wind from surface to 1000m (Fioletov et al., 2017; Dammers et al., 2019). Figure 2 quantitatively compares the wind speeds of these four options using ERA5 data sampled at OMI level 2 observations within the Po Valley





air basin boundary (see Figure 1) from October 2004 to February 2021. These four wind speeds show strong linear correlation with stronger winds when higher altitudes are involved. The surface-1000m wind speed is almost twice as strong as the 10m wind, whereas the two intermediate options, the 100m wind and surface-500m wind are similar with a difference of 13%. The wind directions among those four options show much larger discrepancy, but only the wind speeds will be used in this study.

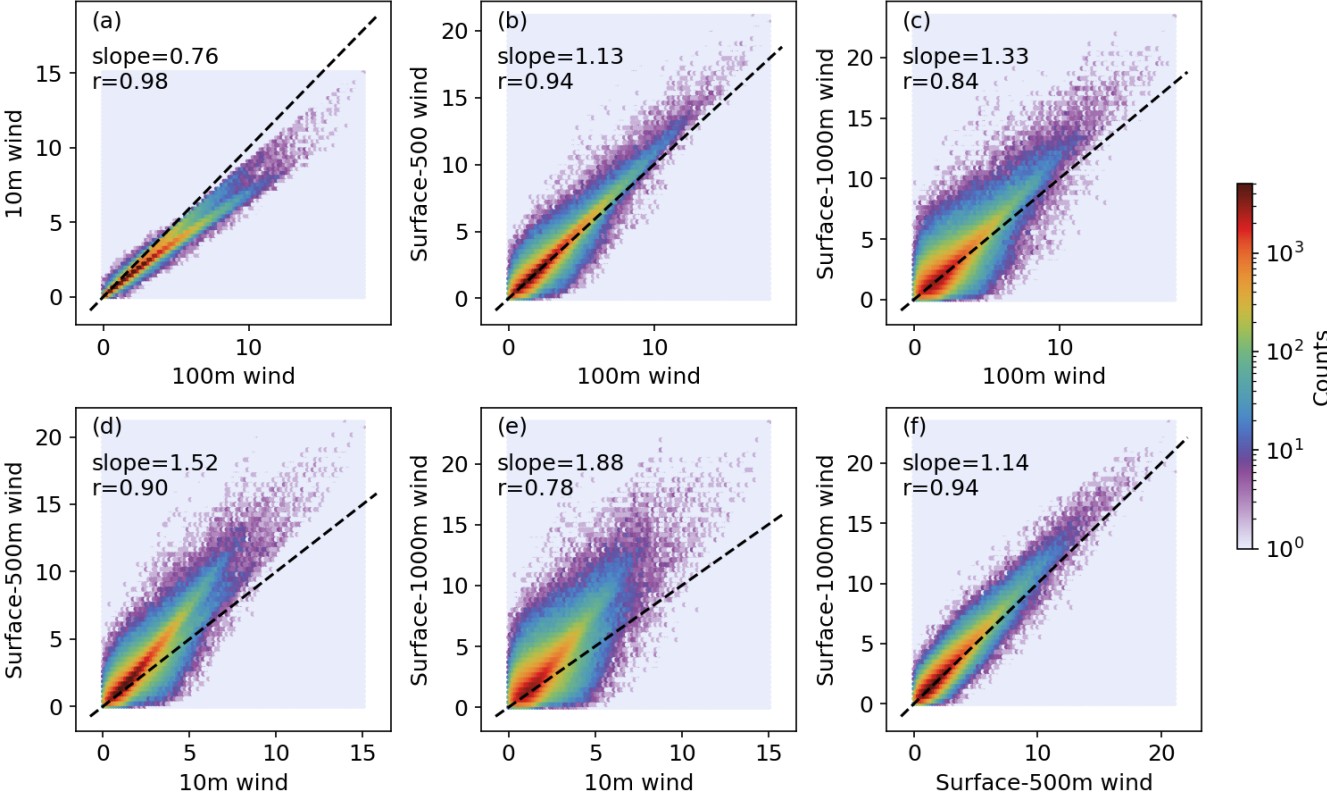

**Figure 2.** Correlations between speeds of 100m wind and 10m wind (a), 100m wind and surface-500m wind (b), 100m wind and surface-1000m wind (c), 10m wind and surface 500m wind (d), 10m wind and surface-1000m wind (e), and surface-500m wind and surface-1000m wind (f). Wind data are from ERA5 meteorology sampled at valid OMI NO$_2$ observation locations in the Po Valley air basin in 2004–2021. The slopes labeled in the plot is from orthogonal regression, and $r$ is correlation coefficient. Unit is m s$^{-1}$.

## 3  Methods

### 3.1  Construction of column-wind speed relationships by physical oversampling

A key step to estimate NO$_x$ emission from the observed NO$_2$ TVCDs is to construct the column-wind speed relationship by averaging column amounts over a range of wind speed intervals. Physical oversampling (Sun et al., 2018) provides a flexible way to spatiotemporally average satellite data with proper weighting and slice the data under different environmental conditions





(e.g., wind speed). The averaged NO$_2$ TVCD ($\langle \Omega \rangle$) given sets of filtering criteria with respect to space ($s$), time ($t$), and other

level 2 parameters ($p$) can be calculated as:

$$\langle \Omega \rangle (s,t,p) = \frac{\sum_{j \in s} \sum_{i \in t,p} w_{i,j} \Omega_i}{\sum_{j \in s} \sum_{i \in t,p} w_{i,j}}. \tag{1}$$

Here $j$ is the index of each level 3 grid cell at $0.01°$ resolution, and $j \in s$ includes all grid cells satisfying the spatial aggregation

criterion $s$ (e.g., within the boundary of an air basin). $\Omega_i$ is NO$_2$ TVCD retrieved at level 2 pixel $i$. $i \in t,p$ keeps only level 2

pixels satisfying time filtering criteria (e.g., within a calendar month) and parameter filtering criteria (e.g., wind speed at the

level 2 pixel within a certain interval). $w_{i,j}$ is the weight of level 2 pixel $i$ at level 3 grid cell $j$ and depends on the spatial

response of pixel $i$ at grid cell $j$ as well as the retrieval uncertainty at pixel $i$ (Zhu et al., 2017; Sun et al., 2018).

The column-wind speed relationship for an air basin over a certain time interval is an array of averaged NO$_2$ TVCDs over

different wind speed intervals (every $0.5$ m s$^{-1}$ in this study):

$\langle \boldsymbol{\Omega} \rangle = [\langle \Omega \rangle (0 \text{ m s}^{-1} \leq W < 0.5 \text{ m s}^{-1}), \langle \Omega \rangle (0.5 \text{ m s}^{-1} \leq W < 1.0 \text{ m s}^{-1}), \cdots ],$ (2)

where $W$ is the horizontal wind speed that is interpolated at level 2 pixels and representative of horizontal advection. The four

wind speed options shown in Figure 2 are tested in this study. Figure 3 shows the column-wind speed relationships for OMI

and TROPOMI over the Po Valley in December 2018–November 2020 grouped into four seasons. TROPOMI provides 2–3

times more coverage than OMI, as indicated by the dot sizes, but $\sim$50 times more valid level 2 pixels due to much finer spatial

resolution, as labeled in the legends.

### 3.2 Conceptual model of column-wind speed relationships

The emission rate over an air basin $Q$ can be linked to the basin-average column amount through a box model

$$\langle \boldsymbol{\Omega} \rangle = \frac{Q}{\phi A \left( \frac{1}{\boldsymbol{\tau_d}} + \frac{1}{\tau_c} \right)}, \tag{3}$$

where boldface symbols indicate vectors. The averaged NO$_2$ TVCD $\langle \boldsymbol{\Omega} \rangle$ and dynamic lifetime $\boldsymbol{\tau_d}$ are both vectors resolved

over a range of wind speeds $\boldsymbol{W}$, $\phi = 1.32$ is the NO$_x$/NO$_2$ ratio representative of cloud-free mid-day conditions in polluted air

mass (Beirle et al., 2019), $A$ is the air basin area, and $\tau_c$ is the NO$_x$ chemical lifetime. The NO$_x$ chemical lifetime may vary

with wind speed depending on complicated nonlinear chemistry (Valin et al., 2013), so the scalar $\tau_c$ here should be considered

as the average value over the wind speed range. We further simplify the dynamic lifetime dimensionally as the ratio between

wind speed and the horizontal length scale of the air basin $L$:

$$\boldsymbol{\tau_d} = \frac{L}{\boldsymbol{W}}. \tag{4}$$

This implicitly assumes that the horizontal wind efficiently ventilates pollution away from the air basin, which is invalid at low

wind speed. We thus limit our analysis over moderate wind speeds as will be shown in Figure 3. Then the conceptual model of





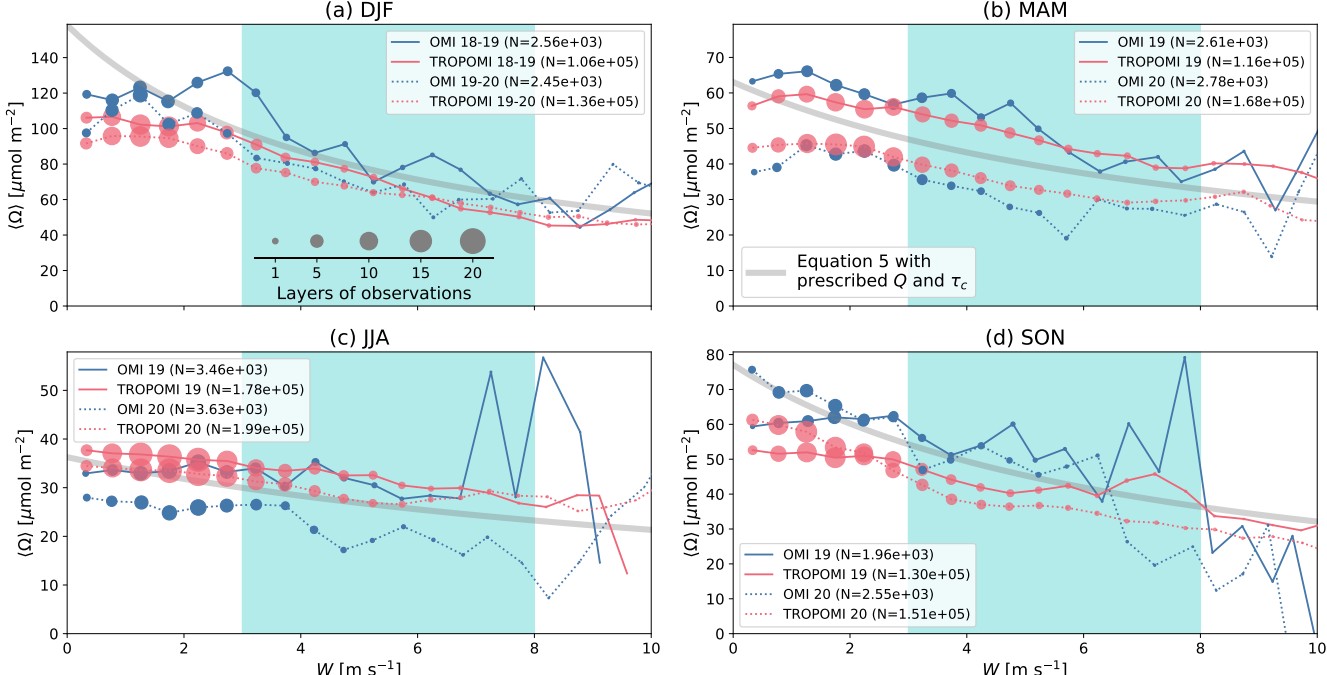

**Figure 3.** Relationships between OMI (blue) and TROPOMI (red) NO$_2$ TVCDs and wind speeds in December, January, February (DJF, a), March, April, May (MAM, b), June, July, August (JJA, c), and September, October, and November (SON, d). Data are shown as solid lines for 2019 (including December 2018) and dotted lines for 2020 (including December 2019). Layers of level 2 observation coverage are indicated by dot sizes. $N$ in the legends denotes total number of level 2 pixels used in each column-wind speed relationship. Thick gray lines show the behaviors of Eq. 5 using prescribed NO$_x$ emission rate ($Q$) of 240, 170, 160, and 170 $\mu$mol m$^{-2}$ and chemical lifetime ($\tau_c$) of 16, 9, 5.5, and 11 hours for the four seasons. Modest wind range of 3–8 m s$^{-1}$ is highlighted by cyan shade.

column-wind speed relationship can be written as

$$\langle \mathbf{\Omega} \rangle = \frac{Q}{\phi A \left( \dfrac{W}{L} + \dfrac{1}{\tau_c} \right)}, \tag{5}$$

Van Damme et al. (2018) and de Foy et al. (2015) have applied such box models to estimate short-lived NH$_3$ and NO$_x$ emission rates by prescribing their chemical lifetimes. The dynamic lifetime was neglected (Van Damme et al., 2018) or calculated as the ratio between near-surface wind speed and half edge length of the square box (de Foy et al., 2015). Similar box models have been also used to infer area-integrated CH$_4$ emission rates from column observations (Buchwitz et al., 2017; Varon et al., 2018). The chemical lifetime of CH$_4$ is negligible, and the dynamic lifetime was constrained by CTM simulations in these

studies. Considering that the four wind options described in Section 2.3 (10m, 100m, surface-500m, and surface-1000m) give different yet strongly correlated wind speed values (Figure 2), we expect different $L$ values are needed for those wind options. End-to-end emission rate estimates are performed using those four wind speed options with a range of $L$ values in Section 4.1.



We found that using 100m wind and $L = 280$ km for the Po Valley air basin (close to $\sqrt{A} = 257$ km) gives emission rate estimates that are most consistent with the JPL chemical reanalysis, which is considered to contain the smallest bias due to high level of observational constraints. This is deemed as a calibration for the dynamic lifetime and is specific to the Po Valley air basin.

The behaviour of Eq. 5 is shown in Figure 3 as gray lines with prescribed emission rate $Q$ and chemical lifetime $\tau_c$ values for each season. Equation 5 implies that the column abundance should monotonously decrease with wind speed and, for the same chemical lifetime, scales with emission rate. When the chemical lifetime gets shorter, the $1/\tau_c$ term becomes larger relative to the dynamic lifetime term $W/L$, and hence the column abundance becomes a weaker function of wind speed. This is demonstrated by the fact that the $NO_2$ TVCDs decrease more rapidly with stronger wind in winter, indicating a longer $NO_x$ chemical lifetime. The overall higher levels of $NO_2$ TVCDs in winter result from the combined effects of longer chemical lifetime and stronger emissions (see Section 4.3 for the seasonality of emission rates derived from this study as well as other top-down and bottom-up inventories).

As shown in Figure 3, the observed column-wind speed relationship deviates from Eq. 5 at lower and upper limits of wind speed. The simple parameterization of dynamic lifetime by $L/W$ assumes that the ventilation of the air basin is driven by horizontal advection, which is not valid when the basin air mass is stagnant. This is supported by the flattening of column-wind speed relationships at low wind speeds. At high wind speed, the number of valid observations rapidly decreases, leading to excessive noise. Therefore, we restrict our analysis to a moderate wind speed range of 3–8 m s$^{-1}$, as indicated by the shaded areas in cyan color in Figure 3.

### 3.3 Retrieving emission rate and chemical lifetime from column-wind speed relationships

As shown in Eq. 5, $\langle \Omega \rangle$ and $W$ are vectors with elements separated by wind speeds, so we may directly fit Eq. 5 to the observed column-wind speed relationships and simultaneously obtain emission rate $Q$ and chemical lifetime $\tau_c$. However, the information of $\tau_c$ mainly comes from the flatness of the observed column-wind speed relationship, and thus the fitted $\tau_c$ is highly sensitive to observational noise. Because $Q$ and $\tau_c$ are strongly anti-correlated, the error in $\tau_c$ is efficiently propagated to the fitted $Q$. For example, the spikes in the observed OMI column-wind speed relationships in Figure 3c-d would result in unphysically low chemical lifetime and unrealistically high emission rate without proper regularization. To reliably retrieve $Q$ for each calendar month throughout the OMI and TROPOMI record, we build a monthly climatology of $\tau_c$ from aggregated observation data and use it as prior information in a Bayesian optimal estimation framework (Rodgers, 2000; Brasseur and Jacob, 2017). The steps are summarized below, followed by described description in this section.

1. The monthly column-wind speed relationships are aggregated into 12 months for all the years ("climatological months" hereafter, in contrast to calendar months), separately for OMI (2004-2021) and TROPOMI (2018-2021). $\tau_c$ and $Q$ are then fitted from the column-wind speed relationship of each climatological month.

2. The fitted $\tau_c$ values in the previous step are used as prior constraint in a Bayesian inversion to optimally estimate $NO_x$ chemical lifetimes in the 12 climatological months.





3. The optimally estimated $\tau_c$ climatology is used as prior constraint to retrieve emission rate $Q$ and $\tau_c$ for each calendar month, separately for OMI and TROPOMI.

### 3.3.1 Constructing and fitting climatological column-wind speed relationships

The column-wind speed relationship of each climatological month is averaged from 3-month windows in all available years. For example, the climatological month June is averaged from May–July in 2005–2020 for OMI and 2018–2020 for TROPOMI. Although each climatological column-wind speed relationship is averaged from a significant number of calendar months (48–51 for OMI and 7–9 for TROPOMI), unregularized nonlinear fitting of $Q$ and $\tau_c$ is still highly unstable. Figure 4 shows the independent fitting of the column-wind speed relationships for each climatological month for OMI (a-b) and TROPOMI (c-d) as black symbols. The gray symbols show 100 bootstrap realizations for each climatological month, where the calendar months used for averaging are selected randomly with replacement in each realization. This bootstrapping is necessary for realistic error estimation, as the fitting errors are substantially biased low due to strong anti-correlation of fitted parameters. The uncertainties are very large in certain climatological months. Some climatological months (April and September for OMI and August–October for TROPOMI) are characterized by nonphysically high emission rate and low chemical lifetime, whereas others (January and February for OMI) are subject to spurious high chemical lifetime. Those originate from irregular features on the column-wind speed relationship (observable in Figure 3) and tend to be more significant when satellite coverage is low.

We additionally remove "outlier" calendar months that would significantly alter the fitted $\tau_c$ and $Q$ from the climatological column-wind speed relationship. These outlier months are often characterized by anomalously high $NO_2$ TVCDs over a few wind speed bins. For each calendar month, the corresponding climatological month is processed twice, with and without that calendar month included in the averaging. The differences of the fitted $Q$ and $\tau_c$ the climatologial month with and without a specific calendar month are displayed in Figure 5. The calendar month is excluded as an outlier if the absolute value of its impact on the climatological $Q$ is larger than 70 mol s$^{-1}$ or the absolute value of its impact on the climatological $\tau_c$ is larger than 1.5 h. The long record of OMI enables a second round of outlier removal, where the climatology is averaged from a single month (instead of 3-month window). It is impossible to do that for TROPOMI as one climatological month would only have 2–3 calendar month to average from. In this round, the max $Q$ difference with/without including a calendar month is still 70 mol s$^{-1}$, but the max $\tau_c$ difference is relaxed to 5 h. The excluded calendar months are highlighted by red dots in Figure 5. More winter months are excluded due to lower coverage and consequently noisier column-wind speed relationship. 53% of winter calendar months in the OMI record are excluded, while the overall removal rate is 30%.

After identifying and excluding the outlier calendar months, the climatological column-wind speed relationships are finalized and the climatology of emission rates and chemical lifetimes are fitted again. The results are shown by Figure 6. The fitting quality is significantly improved, as indicated by the reduced variation of bootstrap realizations.

### 3.3.2 Optimal estimation of climatological chemical lifetime

The climatological $NO_x$ chemical lifetimes fitted from the previous step are still unsatisfactory due to remaining large errors and correlation between fitted emission rates and chemical lifetimes. For instance, the OMI-based chemical lifetimes in

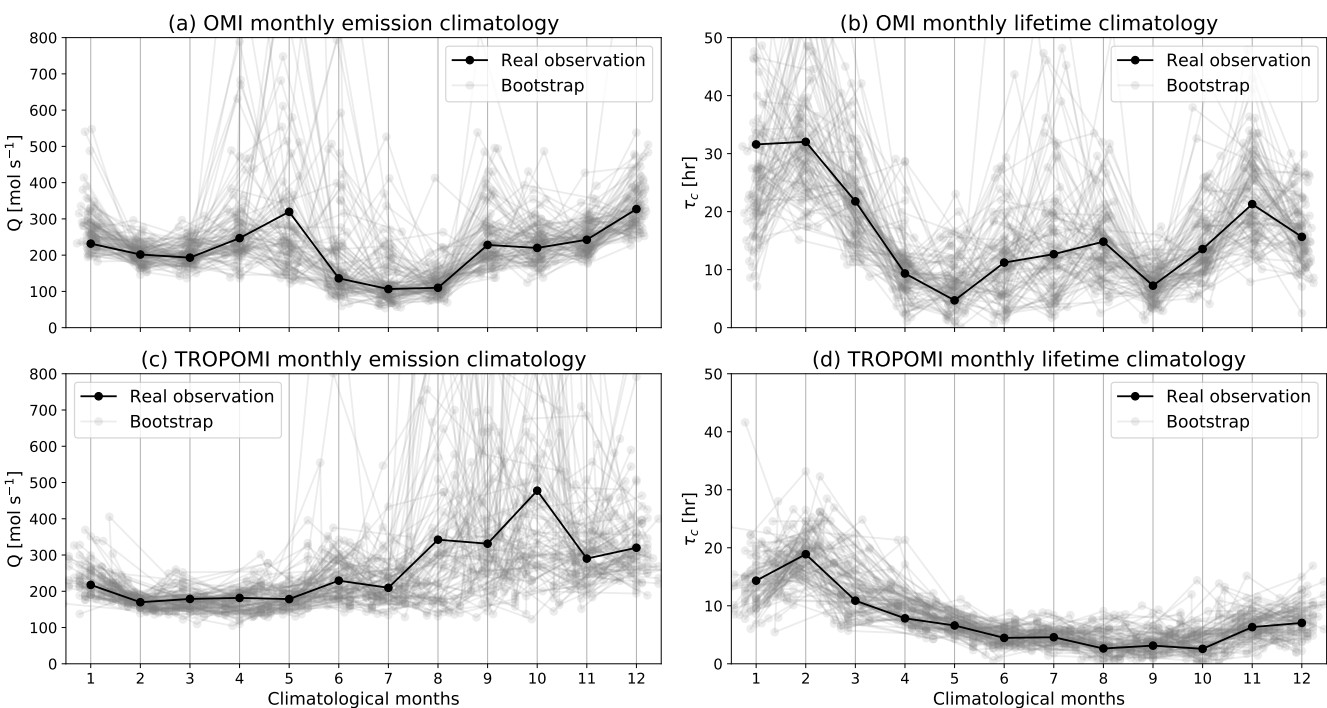

**Figure 4.** Fitting of the column-wind speed relationships for each climatological month for OMI (a-b) and TROPOMI (c-d). The black symbols are fitted from real observed data, and the gray symbols are bootstrapping realizations.

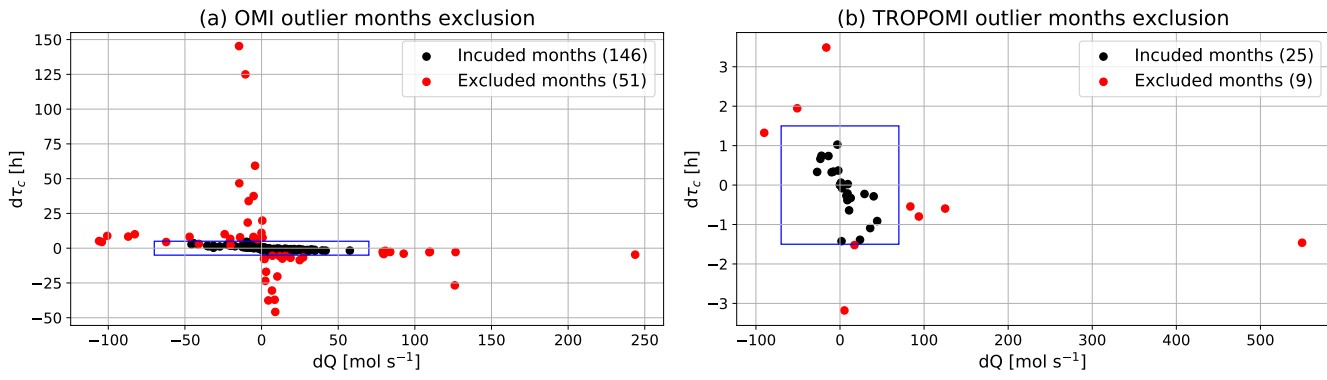

**Figure 5.** Exclusion of outlier months (red) for OMI (a) and TROPOMI. Each dot locates the differences of $Q$ and $\tau_c$ from the corresponding climatological month fitting with and without a specific calendar month. The blue boxes show the boundaries delineating the maximum tolerated $Q$ and $\tau_c$ influences from each calendar month to their corresponding climatological month.





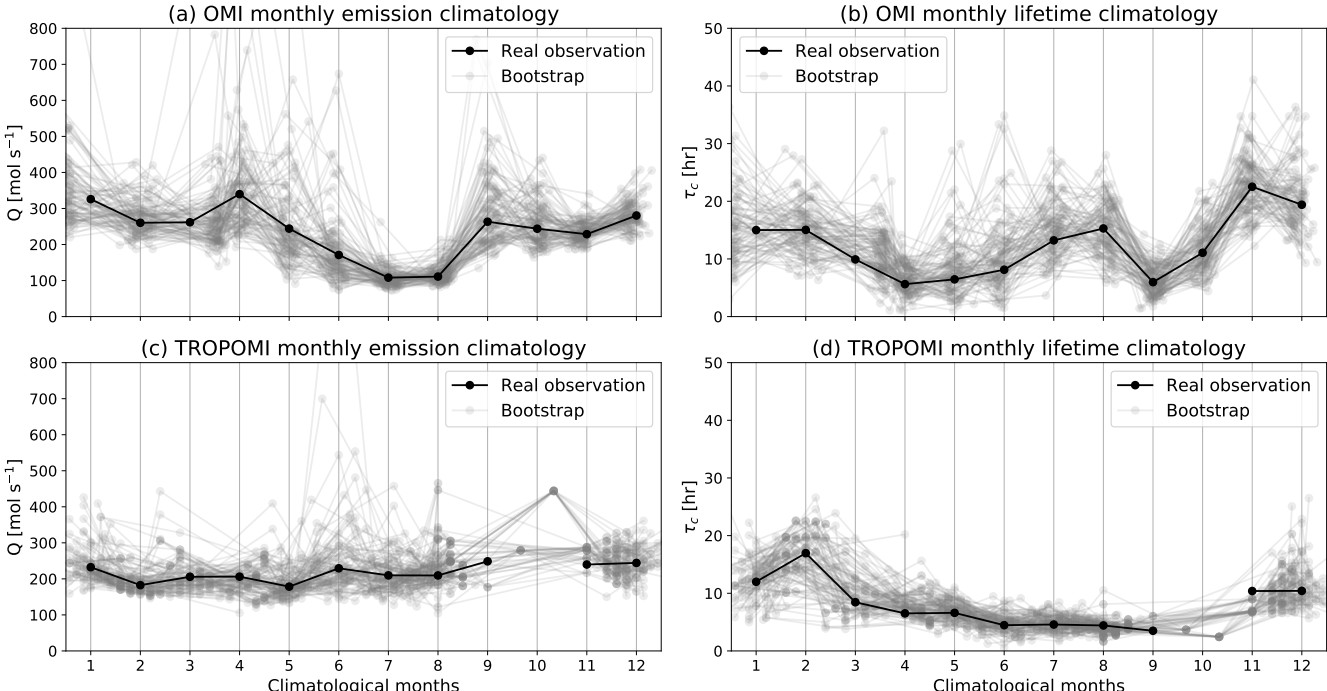

**Figure 6.** Similar to Figure 4, but after outlier months exclusion shown in Figure 5.

climatological months April and September are unrealistically shorter than the summer months (Figure 6b), which is incon-
sistent with the TROPOMI values (Figure 6d) and corresponds to suspiciously high emission rates in those two climatological
months (Figure 6a). To further improve the climatology estimates, we incorporate the a priori information that the climatology
should vary smoothly over the year through a Bayesian optimal estimation. The regularization from the optimal estimation will
effectively suppress noise in the observed column-wind speed relationship.

In this optimal estimation setup, the 12 climatological column-wind relationships are concatenated into a single observation
vector, and the 12 climatological chemical lifetimes and emission rates are retrieved simultaneously as a 24-element state
vector. The fitted OMI- and TROPOMI-based $\tau_c$ values with outlier calendar months removed (black symbols in Figure 6b
and d) are averaged together and smoothed by a first-order Savizky-Golay filter with a 3-month window (Savitzky and Golay,
1964). This smoothed curve is used as the prior values of chemical lifetimes for both OMI and TROPOMI. The prior value
for the emission rates is a constant $260 \ \mathrm{mol \ s^{-1}}$ for all climatological months. The prior error standard deviation is loosely set
at $150\%$ for both $Q$ and $\tau_c$, and a time correlation scale of 1.5 month is assumed within the lifetime terms and the emission
rate terms in the prior error covariance matrix. The model-observation mismatch error depends on satellite retrieval error,
the representativeness of satellite observation in the air basin, and the chaotic nature of atmospheric motion. Little is known
about the last two sources of error except that longer averaging time may reduce them, so we simplify the model-observation
mismatch error as a single regularization factor $\lambda$ that presents its overall variance. Optimal $\lambda$ values are determined separately





225    for OMI and TROPOMI by balancing the norm of fitting residuals and the norm of prior error-weighted deviation of the
solution to the prior using the L-curve (Hansen and O'Leary, 1993). Details in the optimal estimation set up are provided in
Appendix B.

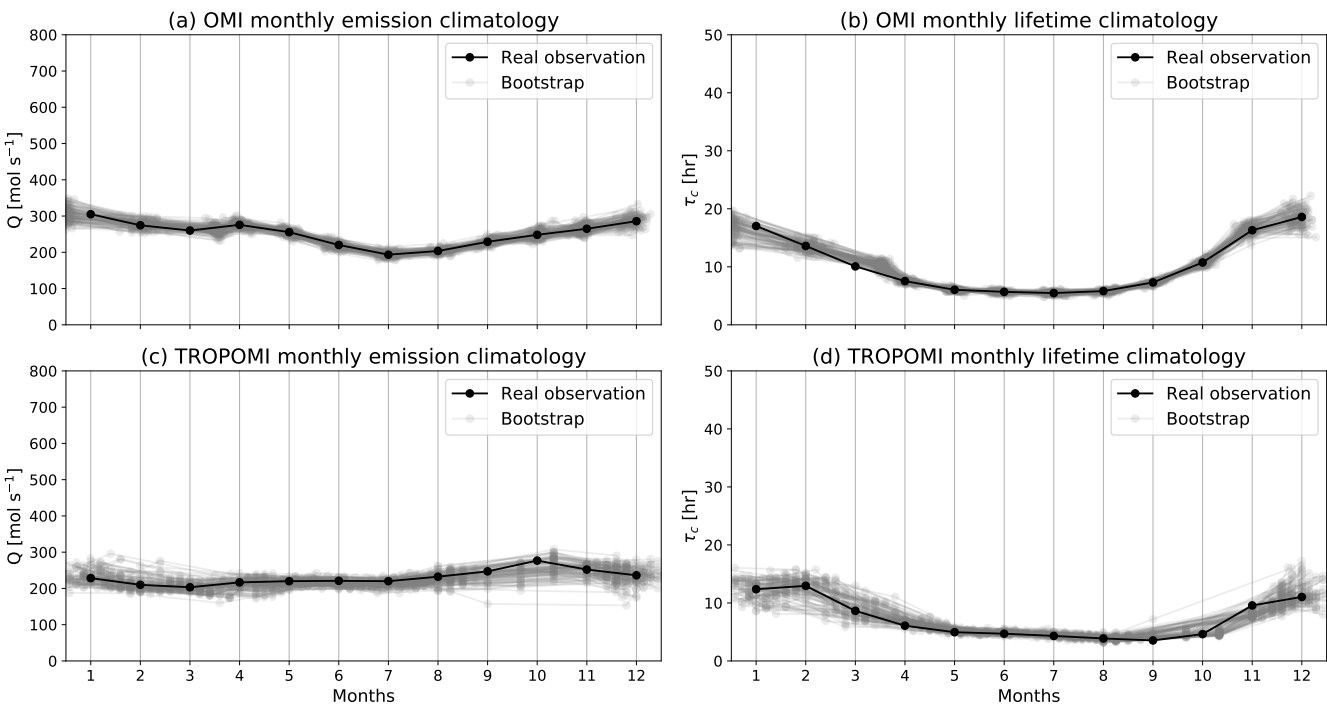

**Figure 7.** Similar to Figure 4 and Figure 6, but using Bayesian optimal estimation incorporating prior knowledge that the climatological
emission rates and lifetimes should vary smoothly.

Figure 7 shows the posterior climatological emission rates $Q$ (a and c) and chemical lifetimes $\tau_c$ (b and d) optimally estimated
using OMI (a and b) and TROPOMI (c and d) column-wind speed relationships. After taking into account the correlations
230    between climatological months via Bayesian optimal estimation, the errors are markedly reduced compared with individual
climatological month fittings shown in Figure 4 and Figure 6. The climatological emission rates estimated from OMI data are
higher than TROPOMI, because overall the OMI record covers more early years (2004–2021) than TROPOMI (2018–2021),
and the emission rate has been decreasing (see Section 4.3). The posterior climatological chemical lifetimes will be discussed
in Section 4.2.

235    **3.3.3    Optimal estimation of emission rates and chemical lifetimes for all calendar months**

Finally, the monthly $NO_x$ emission rate and chemical lifetime are retrieved from the column-wind speed relationships of all
calendar months simultaneously in an optimal estimation algorithm (see Appendix B for technical details). The prior values
of monthly $\tau_c$ are taken from the OMI-based $\tau_c$ climatology due to its overall higher quality and longer temporal coverage





(see Figure 7b and d and further discussion in Section 4.2). In other words, the OMI-based posterior chemical lifetimes in the
12 climatological months are used as the prior chemical lifetime in each calendar month for both OMI and TROPOMI. The
prior error of calendar month $\tau_c$ is assumed to be 30% and auto-correlated with an inter-annual time scale of 1.5 years and
an intra-annual time scale of 1.5 months. This prior regularization to the $\tau_c$ terms is instrumental in the successful retrieval of
emission rate $Q$. The prior values of monthly $Q$ are estimated from an exponential function fitted from the annually averaged
JPL chemical reanalysis emission rates, and 100% prior errors are used. No error correlations are assumed among the $Q$ terms
and between $Q$ and $\tau_c$ terms. This configuration maximizes the information content of emission rates $Q$ from observations
while suppressing excessive noise in the results.

## 4 Results

### 4.1 Selection of air basin length scale

Equation 5 expresses the dynamic lifetime of NO$_x$ in an air basin dimensionally as the ratio between a length scale $L$ and wind
speed. To assess the uncertainties induced by such simplification, we conduct sensitivity studies using end-to-end emission
rate and chemical lifetime estimations described in Section 3.3 by switching wind speed options described in Section 2.3 and
varying the prescribed values for $L$. The resultant OMI-based emission rates are compared with total surface NO$_x$ emission
rates from the JPL chemical reanalysis. We choose OMI-based emission rates due to its long-term consistency and large overlap
with the JPL chemical reanalysis. The combined wind speed option and $L$ value that gives the closest agreement with the JPL
chemical reanalysis monthly emission rate is selected, as the overall accuracy of the the JPL chemical reanalysis is constrained
by multiple observation datasets.

Figure 8a shows the root-mean-square error (RMSE) between the OMI-based emission rates and corresponding JPL chem-
ical reanalysis values in 2005–2019, and Figure 8b compares the temporally averaged emission rates. The optimal $L$ value,
characterized by the lowest RMSE and the matching of temporal mean emission rates to the JPL mean value, increases in the
order of 10m, 100m, surface-500m, and surface-1000m wind, consistent with the overall magnitude of those four wind options.
As shown by Figure 2, those four wind speeds are well linearly correlated. Therefore, the optimal $L$ value scales with the wind
strengths and partially "absorb" the systematic differences between wind speed options. We choose 100m wind due to its low
optimal RMSE and better representation of horizontal advection than the 10m wind. The basin length scale $L$ is selected to be
280 km, similar to the square root of the air basin area (257 km). One should note that this analysis is specific to the Po Valley
air basin and should be repeated before applying such framework to other source regions.

### 4.2 NO$_x$ chemical lifetimes

The optimally estimated climatological chemical lifetimes, which are already shown in Figure 7b and d are replotted in Figure 9
to emphasize the confidence intervals and the prior values that is common for OMI and TROPOMI. The TROPOMI-based
chemical lifetime estimates are consistently lower than the OMI-based values, but the error bars overlap in climatological

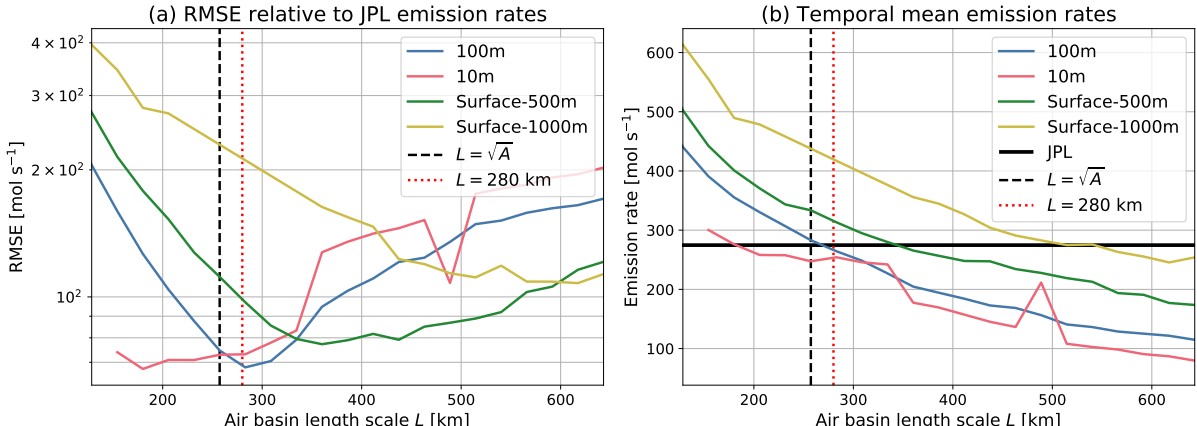

**Figure 8.** (a) Root-mean-square error (RMSE) of OMI-based emission rates relative to the monthly JPL chemical reanalysis emission rates in 2005–2019 when using 10m, 100m, surface-500m, and surface-1000m wind speeds as $W$ in Eq. 5 and a range of $L$ values. (b) Comparison of the temporal mean emission rates estimated from those wind and length scale options with the mean JPL chemical reanalysis emission rate. Square root of the air basin area is shown as the black vertical dashed line, and the selected air basin length scale (280 km) is shown as the red vertical dotted line.

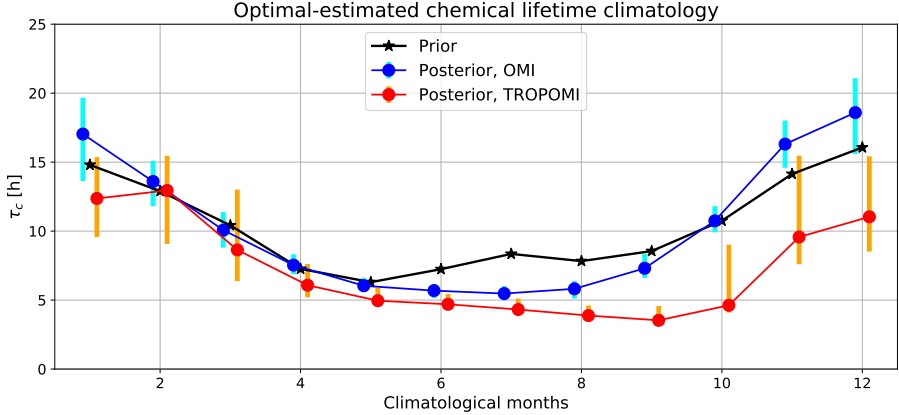

**Figure 9.** Prior (black) and posterior (blue for OMI and red for TROPOMI) climatological chemical lifetimes from optimal estimation. The prior chemical lifetimes are based on nonlinear fitting to the climatological column-wind speed relationships. The error bars indicate $95\%$ confidence intervals by bootstrapping the calendar months used to construct each climatological column-wind speed relationship.





months January–July, indicating that the differences are not significant. Because the OMI climatology spans 2004–2021 while TROPOMI one spans 2018–2021, this difference implies a weak yet notable long-term decrease of $NO_x$ chemical lifetime. This is likely due to the decrease of $NO_x$ emissions (see Figure 11) and consequently the shifting of chemical regimes away from $NO_x$-saturated conditions (Martin et al., 2004). Shifting in summertime $NO_x$ chemical lifetime due to change of $NO_x$ abundance and chemical regimes has been identified in North American cities using OMI observations and an EMG-based approach (Laughner and Cohen, 2019). Model studies indicated similar $NO_x$ chemical lifetime change in polluted regions undergoing decreasing emissions. Using the GEOS-Chem CTM, Silvern et al. (2019) found that the annual mean tropospheric $NO_2$ column lifetime over the contiguous US is 8.1 h in 2005 and 7.7 h in 2017. Shah et al. (2020) simulated $NO_x$ lifetime to be 6.1 h and 27 h in summer and winter in 2012 and 5.9 h and 21 h in summer and winter 2017 using GEOS-Chem in China.

The TROPOMI-based climatological chemical lifetimes are suspiciously low after September. As the $NO_x$ sinks are driven by ambient temperature and solar radiation, we do not expect lower chemical lifetimes in September–October than June–July. This anomaly likely results from abnormal TROPOMI column-wind speed relationships characterized by high $NO_2$ TVCDs in a few wind speed bins. Although the individual monthly column-wind speed speed relationship from OMI is noisier than TROPOMI (Figure 3), the much longer OMI record (197 calendar months vs. 34 calendar months for TROPOMI) enables more effective removal of outlier months and retrieval of climatological chemical lifetimes. As such, we focus on the OMI-based chemical lifetime climatology for the following analysis. The $NO_x$ climatological chemical lifetimes are 5–6 h in summer and 15–20 h in winter, generally consistent with CTMs studies that consider $NO_x$ sinks comprehensively (Mijling and Van Der A, 2012; Stavrakou et al., 2013; Silvern et al., 2019; Shah et al., 2020). The summertime $NO_x$ chemical lifetime is also close to or slightly higher than other observational data-driven estimates, mostly through fitting the downwind decay of $NO_2$ plumes (Valin et al., 2013; de Foy et al., 2015; Liu et al., 2016; Goldberg et al., 2019a; Laughner and Cohen, 2019). This is consistent with the modeling verification by de Foy et al. (2014), which found the $NO_x$ chemical lifetime derived from EMG-based approach to be biased low compared to the true lifetimes in the model simulations.

The OMI-based climatological chemical lifetimes in Figure 9 are then used as the prior to derive chemical lifetimes in each calendar month, for both OMI and TROPOMI. The resultant monthly $NO_x$ chemical lifetimes are shown in Figure 10a. Note that the chemical lifetimes in Figure 10a are retrieved from column-wind speed relationships for each calendar month, whereas the chemical lifetimes in Figure 9 are retrieved from column-wind speed relationships for each climatological month. The observational information content of $\tau_c$ for each calendar month, as indicated by the degrees of freedom for signal (DOFS), is only $\sim 0.02$ (Figure 10b). This reflects our trade-off between emission rates and chemical lifetimes by applying relatively strong prior regularization to $\tau_c$ in each calendar month. It also implies that the chemical lifetimes for calendar months are dominated by prior influences from the climatological chemical lifetimes. While the climatological chemical lifetimes are also derived from observations, the lack of observational constraints for the lifetime in each individual calendar month makes them closely resemble the corresponding climtological month values (i.e., the prior) and prevents us from further interpretation of these monthly lifetime values.

The information of retrieved emission rate $Q$ that is gained from observations, indicated by the corresponding DOFS, is however high and close to unity (Figure 10c). This indicates that we can confidently retrieve emission rates from the monthly


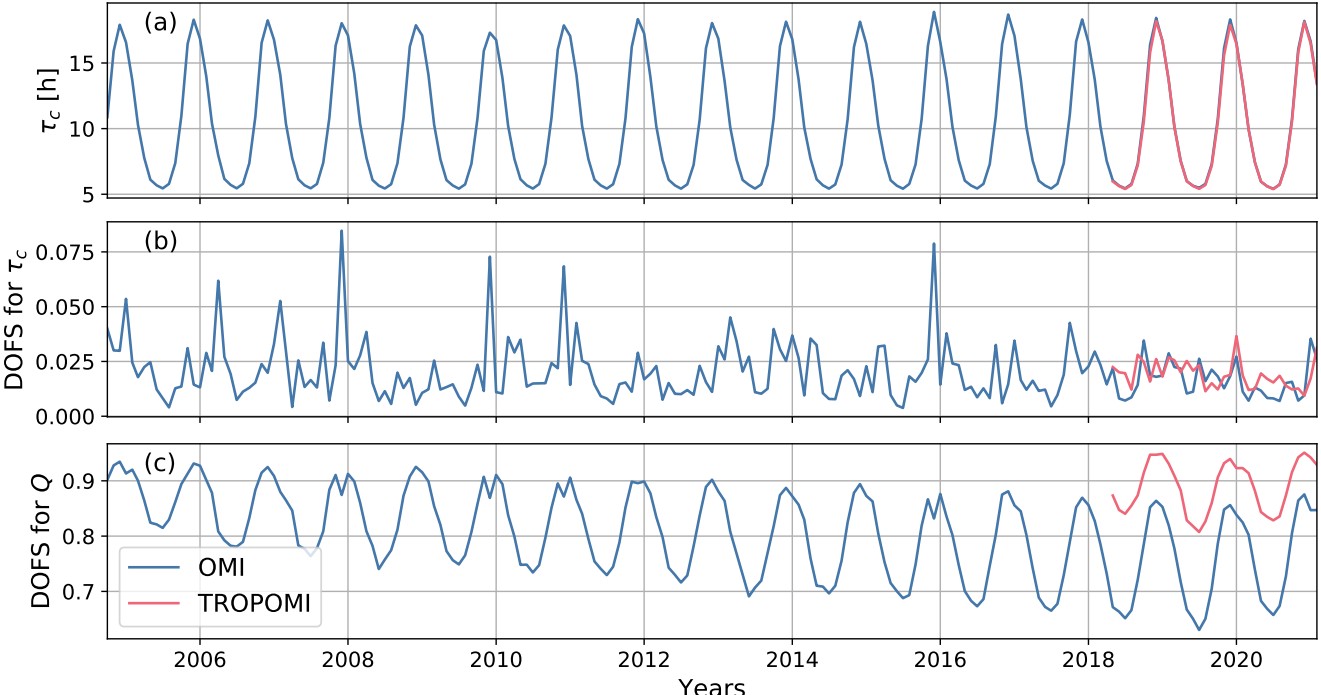

**Figure 10.** Time series of chemical lifetime (a), DOFS for chemical lifetime $\tau_c$ (b), and DOFS for emission rate $Q$ (c) from the calendar month-based optimal estimations using OMI (blue) and TROPOMI (red) monthly data.

column-wind speed relationships. The decaying DOFS for OMI-based emission rates from 2004 to 2021 and the higher DOFS from TROPOMI than OMI are consistent with the instrument performances.

## 4.3 NO$_x$ emission rates

Figure 11 presents the monthly air basin-scale emission rate retrieved from OMI and TROPOMI column-wind speed relationships. The long-term trend and seasonality of OMI-based emission rates match closely to the monthly emission rates from the JPL chemical reanalysis. The emission rates from bottom-up inventories EDGAR, PKUNOx, and CEDS are also shown in Figure 11, where EDGAR is only available as annual average. We use the surface total NO$_x$ emissions from JPL chemical reanalysis, which does not include lightning (1.8% of surface total). According to JPL chemical reanalysis, 96.2% of surface total NO$_x$ emission is anthropogenic. All sectors from CEDS, EDGAR, and PKUNOx are used. Although the PKUNOx and CEDS inventories are monthly, their seasonality differs significantly from the OMI-based and JPL chemical reanalysis values. The inter-annual trends agree reasonably well between bottom-up inventories and top-down emission estimates (JPL chemical analysis and this study) in their overlapping periods, although the emission decrease trends are not as strong in the top-down estimates as in the bottom-up estimates. The TROPOMI-based emission rates show similar variation to OMI and JPL chemical reanalysis, but tend to be lower than OMI in the cold months and higher than OMI in the warm months. The calendar





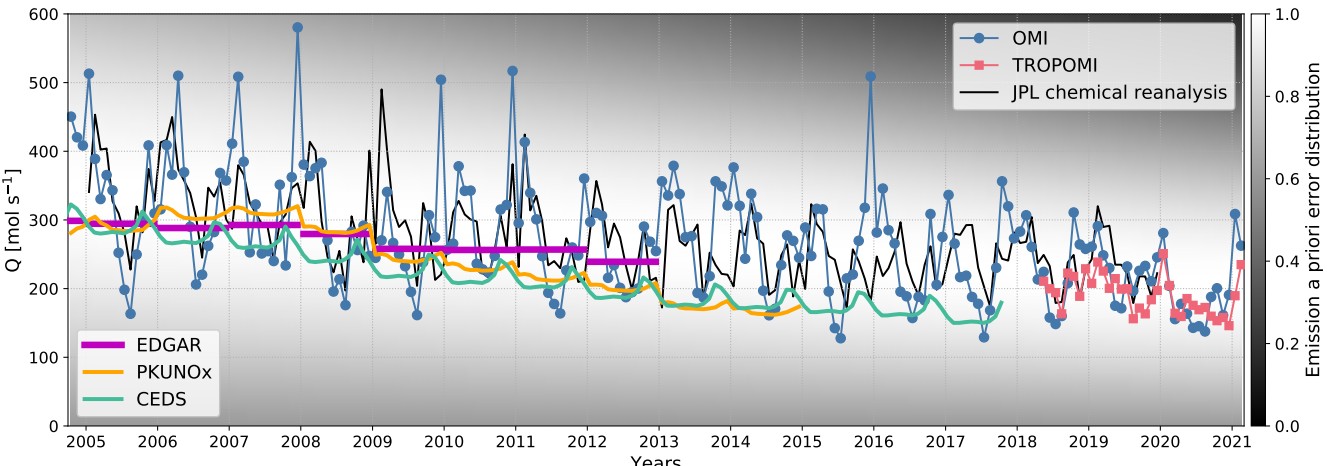

**Figure 11.** Po Valley NO$_x$ emission rates retrieved from OMI (blue circles) and TROPOMI (red squares) column-wind speed relationships in each calendar month. Monthly emission rates calculated from the JPL chemical reanalysis, EDGAR, PKUNOx, and CEDS inventories are shown as black, magenta, yellow, and cyan lines. The background colormap indicates the prior error distributions normalized to peak height of unity for each calendar month.

month chemical lifetimes retrieved from TROPOMI are similar to OMI (Figure 10a), and hence the differences in OMI- and
TROPOMI-based emission rates directly result from differences in their NO$_2$ TVCDs. This is supported by Figure 3; when wind speed is controlled, the OMI TVCDs are higher in cold months while the TROPOMI TVCDs are higher in warm months.

A main advantage of the proposed satellite data-driven framework is to timely quantify rapid emission perturbations. The Po Valley region experienced two major COVID-19 outbreaks, one in February–May and the second one starting from October and ongoing (Dong et al., 2020). Both triggered lockdown measures that are expected to reduce NO$_x$ emission. However, the
quantitative measure of net emission reduction due to the lockdowns is complicated by the long-term decreasing trend and intra-annual variability. For instance, the simple difference between 2020 and 2019 values includes both the pandemic-induced emission changes and the business-as-usual decrease. Leveraging the long and consistent OMI record, we train a statistical model to present the inter- and intra-annual variability using the OMI-based emission rates from January 2010 to December 2019 (yellow shaded region in Figure 12a):

$$Q(t) = \exp\left(\sum_{i=0}^{n_p}\left(c_i t^i\right) + \sum_{j=1}^{n_h}\left(a_j \sin\left(2\pi t j\right) + b_j \cos\left(2\pi t j\right)\right) + e\right), \tag{6}$$

where $t$ is time measured in fractional years, resolved by month, $c$, $a$, and $b$ are model parameters, and $e$ is an error term. The order of polynomial $n_p = 3$ and the number of harmonics $n_h = 5$ are chosen through the Akaike Information Criterion (Akaike, 1974). The fitted model and 95% confidence intervals are estimated using ordinary least squares and displayed in Figure 12a-b. Because the model fitting does not involve data before 2010 and after 2020, the model line over those years are from
extrapolation and characterized by increasingly large uncertainties (i.e., broader confidence intervals) as the range of projection



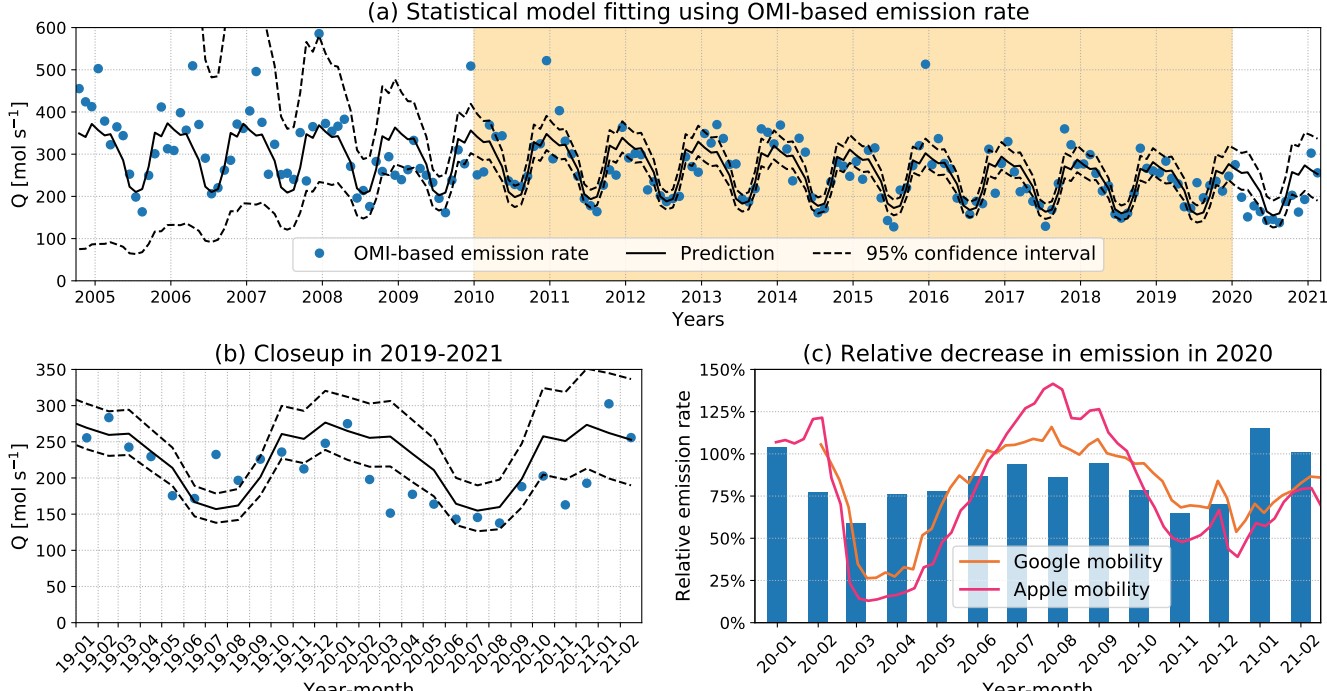

**Figure 12.** (a) The blue dots show the monthly OMI-based emission rates. The black solid and dash lines show the prediction as well as 95% confidence intervals using the model as in Eq. 6. Only data points in 2010–2019 (yellow shade) are used to fit the model. Prediction values outside this range are extrapolation. (b) is similar to (a) but focused on the period after 2019. (c) The bars show real 2020-2021 emission rates relative to the predicted emission rates. The yellow and red line shows Google and Apple mobility indicators.

grows. The well-documented emission perturbation during the 2008–2009 financial crisis (Castellanos and Boersma, 2012) is evident through the discrepancy between model extrapolation and real emission rates (Figure 12a). Similarly, since this statistical model is trained using data before the pandemic, the prediction in 2020 and beyond serves as a business-as-usual baseline. Compared to just using a previous year or multi-year averaged climatology as reference (Goldberg et al., 2020; Liu
et al., 2020; Bauwens et al., 2020), the model prediction incorporates both the long-term trend and seasonality and is less sensitive to noise in monthly estimates. The real emission rates during the pandemic relative to the predicted emission rates are shown in Figure 12c. Significant COVID-19-induced emission reduction started in February and peaked in March at 41%. The emission rate gradually recovered as the first outbreak was under control and reached 85–95% of the pre-existing trajectory in June–September. The emission rate dropped again in October and reached a reduction of 35% relative to the no-pandemic
scenario in November, corresponding to the second outbreak and the subsequent controlling measures. The emission rates in January and February 2021 seem to be back to the expected normal. Overall, the real annual emission of 2020 is estimated to be 20% lower due to the net effect of the COVID-19 pandemic in the Po Valley air basin.





We further correlate the COVID-19-induced $NO_x$ emission changes with the qualitative indicators of human activities estimated by the mobility of Google (Google LLC, 2021) and Apple (Apple, 2021) users. The Google mobility is measured by
the aggregated Google user activity levels at grocery & pharmacy, parks, transit stations, work places, and retail & recreation relative to a baseline period during 3 January–6 February 2020. Google mobility reported for six Italian regions at the Po Valley air basin, including Piedmont, Lombardy, Veneto, Liguria, Emilia-Romagna, and Friuli-Venezia Giulia are averaged. The Apple mobility is measured by Apple user activity levels in driving and transit modes over the entire Italy relative to the baseline on 13 January 2020. Both Google and Apple mobility indicators are in daily native resolution and averaged weekly
to remove day-of-week effects. The result is shown in Figure 12c. The relative $NO_x$ emission changes and the mobility indicators consistently show the "W" shape with the two troughs corresponding to large outbreaks. The impact of the second outbreak was lower than the first one, which is also consistent between the mobility indicators and OMI-based net emission changes. Discrepancies are noted in April 2020 and January 2021, when the mobility indicators stayed low after major control measures, but the $NO_x$ emission recovered quicker. We speculate this to be the impact of industrial $NO_x$ emissions that are not
well represented by the human mobility indicators.

## 5   Conclusions and discussion

We present a satellite-data driven framework to rapidly quantify $NO_x$ emission rates over an air basin, and demonstrate it in Po Valley, Italy. Monthly emission rates and chemical lifetimes of $NO_x$ are retrieved from observed column-wind speed relationships, where the $NO_x$ column abundance is represented by OMI and TROPOMI $NO_2$ TVCD observations, and the
wind speed is obtained from ERA5 reanalysis. To regularize the retrieval, we derive a $NO_x$ chemical lifetime climatology and use it as prior information. The $NO_x$ chemical lifetime is 5–6 h in summer and 15–20 h in winter. Our observation-based emission rate estimates are consistent with top-down and bottom-up inventories and can be quickly updated as the method only depends on satellite and reanalysis data. Leveraging the long and consistent OMI record, a statistical model is trained to predict the business-as-usual trajectory without the pandemic. Compared with this trajectory, the real 2020-2021 emission rates show
two distinctive dips that correspond to tightened COVID-19 control measures and reduced human activities. The overall net $NO_x$ emission reduction due to the COVID-19 pandemic is estimated to be $20\%$ with maximum reduction in March, followed by November.

Only observations under modest wind (3–8 m s$^{-1}$) are used, so there is an implicit assumption that $NO_x$ emissions under modest wind can represent all wind conditions. Since $NO_x$ sources in air basins are mostly anthropogenic, this assumption is
deemed to be valid. In addition, the satellite observations are made in the early afternoon local time, so the retrieved emission rates may not necessarily represent the diurnal mean emission rate. This is a common limitation of all observational data-driven approach, and we note that the overall emission rate level is anchored to the overall emission rate level of JPL chemical reanalysis, which is spatiotemporally complete, through the selection of basin length scale $L$. The uncertainties of the retrieved monthly emission rates may also originate from the systematic biases of $NO_2$ TVCD products, but the relative emission
variations should be insensitive to the observational biases. Updated satellite products (e.g., the version 2 TROPOMI $NO_2$





product to be released in 2021) can be readily adopted. The $NO_x:NO_2$ ratio over the air basin has been fixed at a value of 1.32 (Beirle et al., 2011; de Foy et al., 2015; Liu et al., 2016). The uncertainty of this ratio, up to 20% according to Beirle et al. (2019), will directly propagate to the $NO_x$ emission rate estimate but have limited effect on the relative emission changes. Future work is suggested to better understand its variability. The general framework is not limited to $NO_2/NO_x$ in the Po Valley

air basin, but can be applied to investigating the emissions and lifetimes of other short-lived species in other geographical regions.

*Code availability.* Code relevant to this paper can be found at https://github.com/Kang-Sun-CfA/Oversampling_matlab/, last access: 25 March 2021

*Data availability.* The OMI data are available through NASA GES DISC at https://doi.org/10.5067/Aura/OMI/DATA2017. The TROPOMI

data are available through Copernicus Open Access Hub at https://doi.org/10.5270/S5P-s4ljg54. The ERA5 data are available at https://doi.org/10.24381/cds.adbb2d47. The JPL chemical analysis data are available at https://doi.org/10.25966/9qgv-fe81.

## Appendix A: Pixel-based comparison between OMI and TROPOMI $NO_2$ TVCDs

The TROPOMI retrievals within $\pm 1$ hour from an OMI retrieval are averaged using the relative pixel overlapping area as weight, and only OMI pixels that are $> 80\%$ covered by such TROPOMI pixels are used for comparison. This ensures that

OMI and TROPOMI sample essentially the same air mass, and the $NO_2$ differences reflect the inherent differences of those two products. Figure A1 compares the strictly collocated OMI and TROPOMI retrievals from December 2019 to November 2020. Since the TROPOMI value in each TROPOMI-OMI pair is weight-averaged by 10–90 TROPOMI pixels, its random error is significantly lower than the OMI value, and hence we use the slope in ordinary least square (OLS) regression to represent the OMI/TROPOMI ratio.

Figure A2 compares both the OLS slope and the OMI-TROPOMI NMB. In general, OMI is higher than TROPOMI in the cold season, indicated by slopes larger than unity and NMB larger than zero, whereas TROPOMI is higher in the warm season. The temporal variation of the OLS slope and NMB show only moderate correlation (correlation coefficient $r = 0.54$), indicating that the discrepancy between OMI and TROPOMI is more complicated than an zero-level offset or proportional scaling.

## Appendix B: Details in optimal estimations of $Q$ and $\tau_c$

This appendix section provides technical details on the optimal estimation using column-wind speed relationships over both climatlogical months (Section 3.3.2) and calendar months (Section 3.3.3)



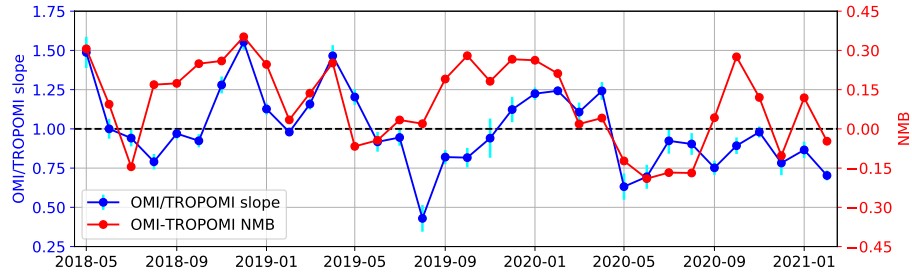

**Figure A1.** Correlation plots between weight-averaged TROPOMI $NO_2$ TVCD at OMI pixel and the corresponding OMI $NO_2$ TVCD. One year data from December 2018 to November 2019 are shown monthly for each panel. Number of collocation pairs ($N$), OMI/TROPOMI slope from OLS regression, and OMI-TROPOMI NMB are shown in each panel. The dashed black line is 1:1.

**Figure A2.** OMI/TROPOMI OLS slope (blue) and OMI-TROPOMI NMB (red) for each month when TROPOMI $NO_2$ TVCD data are available for comparison.

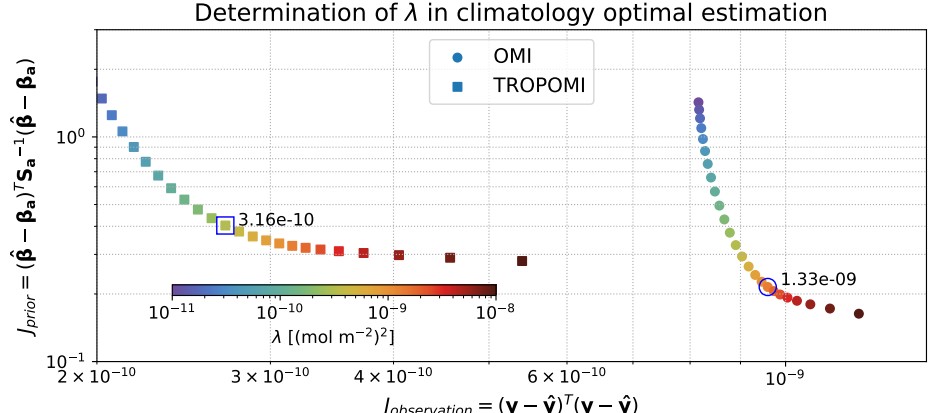

**Figure B1.** L-curve plot of the squared errors of a regularized solution vs. squared residuals for the climatology of OMI (dots) and TROPOMI (squares).

Separately for OMI and TROPMI, the column-wind speed relationship $\langle \Omega \rangle$ vectors are concatenated to a single observational vector ($\mathbf{y}$):

$$\mathbf{y} = [\langle \mathbf{\Omega} \rangle (\text{month1}), \langle \mathbf{\Omega} \rangle (\text{month2}), \cdots]. \tag{B1}$$

There are 12 months for the climatology retrieval, 197 months for OMI-based calendar month retrieval, and 34 months for TROPOMI-based calendar month retrieval. The state vector $\boldsymbol{\beta}$ includes emission rates and chemical lifetimes of all months:

$$\boldsymbol{\beta} = [Q(\text{month1}), Q(\text{month2}), \cdots, \tau_c(\text{month1}), \tau_c(\text{month2}), \cdots]. \tag{B2}$$

The optimal estimation is obtained by iteratively minimizing the cost function

$$J = (\mathbf{y} - \boldsymbol{f}(\boldsymbol{\beta}))^T \lambda^{-1} (\mathbf{y} - \boldsymbol{f}(\boldsymbol{\beta})) + (\boldsymbol{\beta} - \boldsymbol{\beta_a})^T \mathbf{S_a}^{-1} (\boldsymbol{\beta} - \boldsymbol{\beta_a}). \tag{B3}$$

Here $\boldsymbol{f}(\boldsymbol{\beta})$ is the forward model by concatenating Eq. 5 for each month, $\boldsymbol{\beta_a}$ is the prior vector, and $\boldsymbol{S_a}$ is the prior error covariance matrix. In the optimal estimations applied in this study, the strength of prior regularization is controlled by a single factor $\lambda$. Lower $\lambda$ value, or weaker regularization, leads to smaller residuals but larger deviation from the prior; higher $\lambda$ value, or stronger regularization, leads to smaller deviation from the prior but larger residuals. We select $\lambda$ values for OMI/TROPOMI and climatology/calendar months separately, by finding the maximum curvature point. The corresponding L-curve plots in the $Q/\tau_c$ optimal estimations using column-wind speed relationships averaged to climatological months and in each calendar month are shown in Figure B1 and Figure B2, respectively. The selected $\lambda$ values are labeled in the plots.

The cost function $J$ (Eq. B3) is minimized by a Gauss-Newton approach, where the state vector is updated in each iteration by the following rule:

$$\boldsymbol{\beta_{i+1}} = \boldsymbol{\beta_i} + (\mathbf{S_a}^{-1} + \mathbf{K_i}^T \lambda^{-1} \mathbf{K_i})^{-1} (\mathbf{K_i}^T \lambda^{-1} (\mathbf{y} - \boldsymbol{f}(\boldsymbol{\beta_i})) - \mathbf{S_a}^{-1} (\boldsymbol{\beta_i} - \boldsymbol{\beta_a})). \tag{B4}$$





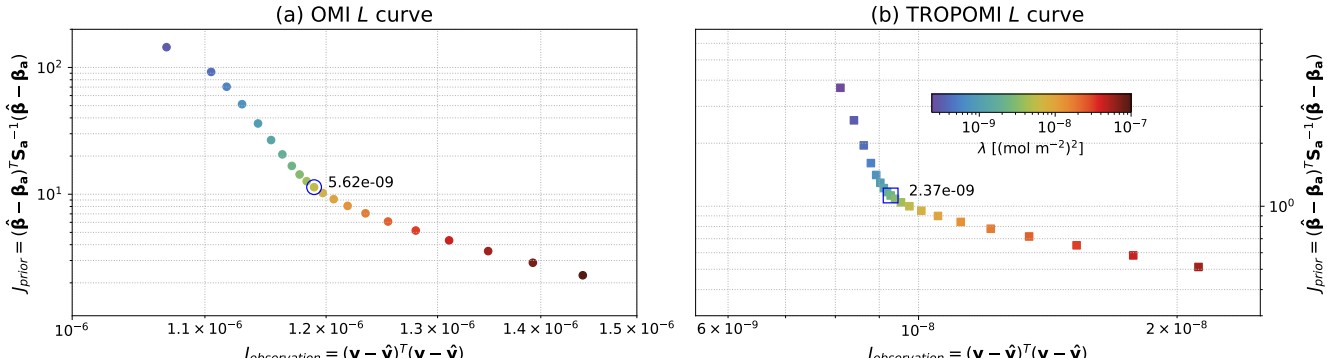

**Figure B2.** Similar to Figure B1 but showing L-curve plots for the optimal estimation for all calendar months for OMI (a) and TROPOMI (b).

Here $\beta_i$ is the state vector estimation in iteration $i$, and $\beta_0 = \beta_a$. $\mathbf{K_i} = \partial f(\beta_i)/\partial \beta_i$ is the Jacobian matrix at iteration $i$. Since the forward model is concatenated from column-wind speed relationship for each month (Eq. 5), and the state vector is concatenated from $Q$ and $\tau_c$ for each month, the Jacobian can be constructed using analytical derivations of Eq. 5:

$$\frac{\partial \langle \mathbf{\Omega} \rangle}{\partial Q} = \frac{1}{\phi A \left( \dfrac{W}{L} + \dfrac{1}{\tau_c} \right)}, \tag{B5}$$


$$\frac{\partial \langle \mathbf{\Omega} \rangle}{\partial \tau_c} = \frac{Q}{\phi A} \frac{1}{\tau_c^2 \left( \dfrac{W}{L} + \dfrac{1}{\tau_c} \right)^2}. \tag{B6}$$

The convergence is determined by comparing the error variance derivative (Bösch et al., 2015)

$$d\sigma_i^2 = (\beta_{i+1} - \beta_i)^T \left( \mathbf{K_i}^T \lambda^{-1} (\mathbf{y} - f(\beta_i)) + \mathbf{S_a}^{-1} (\beta_i - \beta_a) \right) \tag{B7}$$

with a threshold that scales with the number of state vector elements.

After an optimal solution is found, the DOFS for each state vector element ($Q$ or $\tau_c$) is the corresponding diagonal element

of the averaging kernel matrix

$$\mathbf{A} = \mathbf{I} - \left( \mathbf{K}^T \lambda^{-1} \mathbf{K} + \mathbf{S_a}^{-1} \right)^{-1} \mathbf{S_a}^{-1}, \tag{B8}$$

where $\mathbf{K}$ is the Jacobian matrix at the final iteration and $\mathbf{I}$ is an identity matrix with the same dimension as the state vector.

*Author contributions.* KS designed and implemented this study and wrote the paper. LBL helped calculating the top-down and bottom-up inventory emission rates. SJ helped curating satellite data. LBL and SJ contributed to satellite data analysis. DL provided expertise on
atmospheric transport and helped with scientific interpretation and discussion.





*Competing interests.* The authors declare that they have no conflict of interest.

*Acknowledgements.* This research has been supported by the NASA Earth Science Division Rapid Response and Novel Research in Earth Science program (RRNES, Award 80NSSC20K1295) and Atmospheric Composition: Modeling and Analysis (ACMAP, Award 80NSSC19K0988).



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
