# Peer review of "A Satellite Data-Driven Framework to Rapidly Quantify Air Basin-Scale $NO_x$ Emission and Its Application to the Po Valley during the COVID-19 Pandemic"

_Atmospheric Chemistry and Physics, 2021_

## Author Comment (AC1)

Response to Referee #2:

We appreciate the very helpful feedback from the referee. The referee's comments are listed in *italics*, followed by our response in blue. New/modified text in the manuscript is in **bold**.

*This article estimates NOx emissions and lifetimes in the Po Valley of Italy using OMI and TROPOMI satellite retrievals before the current pandemic, and then uses the methods to estimate the reductions in emissions during the pandemic.*

*The authors are to be congratulated for a thorough and methodical analysis and for interesting results and a topical analysis. I am happy to recommend publication.*

*Really my only comment is the reference to DOFS. The acronym should be spelled out in the caption for Fig 10. The method should then be explained and referenced in the main text.*

The DOFS is now spelled out in the caption. The sentences at lines 294-299 of the original manuscript are updated to explain the DOFS:

"**The degrees of freedom for signal (DOFS) of retrieved emission rates and chemical lifetimes, shown by Figure 10b and c, are the diagonal elements of the averaging kernel matrix as given in Appendix B. The DOFS quantifies the number of pieces of information retrieved from observation for a specific state vector element (Rodgers, 2000; Brasseur and Jacob, 2017). The observational information content of $\tau_c$ for each calendar month, as indicated by the DOFS, is only ~0.02 (Figure 10b). This implies that the chemical lifetimes for calendar months are dominated by prior influences from the climatological chemical lifetimes, which reflects our trade-off between emission rates and chemical lifetimes by applying relatively strong prior regularization to $\tau_c$ in each calendar month.**"

*About the seasonal signal in NOx emissions, the authors do note that "future work is suggested to better understand its variability." This is indeed the case – it does seem more likely that the variability is related to the method and the estimates of the lifetimes than to actual variations in emissions. I would recommend some more discussion of this point – either what might cause changes in actual emissions, or how the method can have biases that vary by season.*

The sentence "future work is suggested to better understand its variability" was about the variability of $NO_x:NO_2$ ratio, instead of $NO_x$ emissions. It has been removed as we have updated the manuscript to include observation-based $NO_x:NO_2$ intra-annual variability according to another referee's comments. Sentences at lines 381-383 of the original manuscript are updated to discuss the observation-based $NO_x:NO_2$ ratio:

"**Monthly climatological $NO_x:NO_2$ ratio derived from ground-based observation network is used to convert $NO_2$ abundance to $NO_x$ abundance, which improves upon the fixed value used in previous studies (Beirle et al., 2011, Valin et al., 2013, de Foy et al., 2015, Liu et al., 2016). However, uncertainty remains from contamination of $NO_2$ in-situ measurements (Visser et al., 2019) and the representativeness of surface-based $NO_x:NO_2$ ratio to the column-integrated one due to vicinity to emission sources and local ozone titration. Moreover, long-term trend in $NO_x:NO_2$ may exist as observed in The Netherlands by Zara et al. (2021),**

**although biases in $NO_x$:$NO_2$ have limited impacts on chemical lifetime and relative emission change estimates."**

One more sentence is added after these to discuss the emission variation by season:

**"The seasonal variability of estimated $NO_x$ emission is determined by the seasonal variabilities of $NO_2$ TVCD, chemical lifetime, and $NO_x$:$NO_2$ ratio. We attempt to characterize these variabilites using as much observational data as possible, and yet future investigations are still needed."**

---

## Author Comment (AC2)

Response to Referee #1:

We appreciate the very helpful feedback from the referee. The referee's comments are listed in *italics*, followed by our response in blue. New/modified text in the manuscript is in **bold**.

*The paper by Sun and co-authors presents a study into changes in NOx emissions inferred from OMI and TROPOMI NO2 retrievals between 2005 and 2020 over the Po Valley. The theoretical basis for the inversion approach relies on a textbook-type one box model that is ventilated by wind; the stronger the wind, the more loss of pollution to outflow rather than to chemical loss, and stronger loss overall. The authors exploit this relationship between NO2 levels and wind speed and show it to be captured by OMI and TROPOMI NO2 observations and ECMWF wind data.*

*The method to derive NOx emissions relies on a Bayesian framework with a priori climatologically established relationships between OMI NO2 columns and wind speeds, and furthermore relies heavily on fine-tuning using pre-existing emission estimates. The authors explain this quite well in the manuscript, but they are optimistic in calling their approach 'fast' or capable of providing 'timely' updates to emissions given the sophisticated efforts they require to arrive at the final results. These are nevertheless interesting, and I recommend publication in ACP after the following concerns have been addressed.*

We appreciate the careful review by the referee. The "fine-tuning using pre-existing emission estimates" mentioned above only happens in Section 4.1 "Selection of air basin length scale". We indeed had to tune the air basin length scale to match the overall multi-year emission trend to the JPL chemical reanalysis. However, after this lengths scale (280 km for the Po Valley) is selected, it is considered only a dimensional property of the air basin and independent of time. Then the optimal estimations, with bootstrapping uncertainty quantification, only consume about a minute. The most time-consuming part is actually downloading and subsetting TROPOMI $NO_2$ level 2 data, given their large size.

We update the last sentence in Section 4.1 to emphasize that the air basin length scale is specific to the Po Valley air basin and needs to be re-estimated in other regions:

"**One should note that this length scale is specific to the Po Valley air basin and should be fixed in time. A length scale should be similarly estimated before applying such framework to other source regions.**"

We further update the sentence at line 322 of the original manuscript to

"**Once the air basin length scale is selected (see Section 4.1), the proposed satellite data-driven framework can be used to timely quantify rapid emission perturbations.**"

We have also updated the calendar month-based emission rate estimates from February 2021 (the manuscript was submitted to ACPD on 26 March 2021) to June 2021. Due to inclusion of more data, adding non-constant $NO_x:NO_2$ ratio (see response to the $NO_x:NO_2$ comment below), and changing the number of harmonics ($n_h$ in Eq. 6 of the manuscript) from 5 to 4 in the statistical model (according to the Akaike Information Criterion), the estimate of overall 2020 $NO_x$ emission reduction is updated from 21% to 22%, and the COVID-19-induced emission reductions in March

and November 2020 are updated from 41% to 42% and from 35% to 38%, respectively. Those numbers are updated in the revised manuscript.

One interesting feature revealed by adding March-June 2021 data is the third wave of emission reduction in March-April 2021, which is consistent with Google and Apple mobility indices (and COVID cases in Italy, not shown):

[Figure]

Then sentences in lines 344-346 of the original manuscript are updated to reflect the newly included months:

**"Thereafter, the emission rate dropped twice as of July 2021, reaching reductions of 38% and 39% relative to the no-pandemic scenario in November 2020 and March 2021. These reductions correspond to the second and third outbreaks and the subsequent controlling measures. The emission rates in January-February and May-June 2021 seem to be back to the expected normal, highlighting the evolving nature of pandemic-induced emission perturbations."**

*Major comments*

*P6, Eq. (4): the assumption that wind efficiently ventilates pollution from the Po Valley is rather questionable. Mountains are surrounding much of the valley, so that with winds from the west, the east and the south, air pollution can be expected to accumulate and circulate within the basin, rather than to be transported away from it. The authors should provide a convincing justification for why Eq. (4) would still hold over the Po Valley. To me it seems only true if the winds are coming from the north and blow out pollution to the Mediterranean Sea.*

The following figures show the column-wind speed relationships averaged for each climatological month separately for north, south, west, and east wind directions for OMI (2004-2021):

[Figure]

and TROPOMI (2018-2021):

[Figure]

As one may see from those figures, irregular features do exist in the column-wind speed relationships due to the chaotic nature of atmospheric motion, but northerly wind does not stand out compared to other wind directions. The box model applied in this study represents basin-scale ventilation by wind speed, and is insensitive to wind direction. For this reason, we choose not to filter out data according to wind directions to maximize signal-to-noise ratio in the column-wind speed relationships. We update the sentence at lines 131-132 of the original manuscript (after Eq. 4) to the following:

**"This implicitly assumes that the horizontal wind efficiently ventilates pollution away from the air basin. However, the Po Valley is surrounded by mountains except the east side. Low wind conditions may only circulate air pollution within the basin boundary. We thus limit our analysis over moderate wind speeds as will be shown in Figure 3. We don't find systematic differences in column-wind relationships over different wind directions over moderate wind speeds, so all wind directions are combined to maximize the number of observations."**

*The claim that the method can also be applied to other regions would needs to be put to the test. Given the concern above, it might have been better to test the technique on a polluted area not surrounded by mountain regions, such as the Ruhr Area or the British Midlands.*

As shown by the column-wind speed relationships separated by wind directions plotted in the previous response, winds blowing over mountain boundaries do not seem to differ systematically from those over water boundaries. We are in the process of applying this framework to other source regions and short-lived species and simplify the pipeline. However, we believe that studies for other regions are out of the scope of this manuscript and will report our results in future publications.

*NOx lifetime changes with changing wind speed (Valin et al., 2012; Lorente et al. [2019]), but the method proposed by the authors fits only one lifetime for different wind speed levels. This seems to be an internal inconsistency in the method (Figure 3 and Eq. (5)) and the authors need to quantitatively explain how they circumvent this problem.*

We have emphasized in lines 126-128 that $NO_x$ lifetime varies with wind speed and the retrieved $\tau_c$ should be considered as average lifetime over moderate wind speed range (3-8 m/s). Theoretically, the wind speed dependence of chemical lifetime could be retrieved from the curvature of the column-wind speed relationships as a polynomial between chemical lifetime and wind speed. However, this is not feasible on real data given the high noise level in column-wind speed relationships (Fig. 3 of the manuscript) and low information content in retrieved average chemical lifetime (Fig. 10b of the manuscript), which is just a zero-order polynomial. We acknowledge this by adding the following sentence to line 128:

**"The high noise level in column-wind speed relationships prevents us to obtain further wind speed dependence of chemical lifetime."**

*I'm missing a discussion of the role of soil NOx emissions in the Po Valley. These are likely not fully represented in the bottom-up inventories, but since satellite NO2 measurements observe contributions from all sources, including the sizeable soil NOx source in the agricultural hotspot of the Po Valley (e.g. Visser et al. [2019]), this may well lead to discrepancies between the top-down and bottom-up NOx emission estimates.*

Discussion is added to line 317 of the original manuscript:

**"The JPL chemical reanalysis reports 3.5% of $NO_x$ emission in the Po Valley from soils. However, other top-down studies indicate that the soil emissions may be underestimated in Europe, ranging from 14 to 40% (Visser et al., 2019 and references therein). Since the**

satellites observe emissions from all sources, the discrepancy may also be from missing soil NO$_x$ emissions in bottom-up inventories."

*The other weak point is the assumption that the NOx:NO2 ratio in the polluted boundary layer can be taken as fixed and always have a value of 1.32. The authors should verify the validity of this assumption over the Po Valley, i.e. if there are not trends in the NOx:NO2 ratio following from the reductions in NOx emissions.*

We looked at ground-based NO$_x$:NO$_2$ observations that are available from 2013 to 2021 in the Po Valley. Due to the inconsistent temporal coverage of ground-based NO$_x$ observations, the long-term trend in NO$_x$:NO$_2$ is unclear. However, it is convincing that NO$_x$:NO$_2$ shows an intra-annual cycle with higher values in winter. We therefore implement a seasonal variation in NO$_x$:NO$_2$ ratio and update the monthly emission rate estimates.

We added section 2.4 to describe the ground-based NO$_x$ data:

**"2.4 In-situ NO$_x$ observations**

**We use the ground-based NO$_x$ observations over the Po Valley available from air quality data portal of the European Environment Agency (EEA) to constrain the temporal variation of NO$_x$:NO$_2$ ratio (EEA, 2021). The validated data (E1a) are used for years 2013-2019 and combined with up-to-date data (E2a) for 2020-2021. Only valid hourly data in 13-14 local time with both NO$_2$ and NO$_x$ available are included in the analysis. We include only ground-based observations within OMI level 2 pixels with cloud fraction < 0.3, but the resultant clear-sky vs. all-sky differences are insignificant."**

and section 4.3 to describe the observed NO$_x$:NO$_2$ ratio:

**"4.3 Observational constraints on NO$_x$:NO$_2$ ratio**

**Despite its limited effect on the estimates of NO$_x$ chemical lifetime and relative emission changes, the uncertainty of NO$_x$:NO$_2$ ratio ($\phi$ in Eq. 5) will directly propagate to the NO$_x$ emission rate estimates. We investigate ground-based NO$_x$:NO$_2$ ratio measured at EEA sites as labeled in Figure 11a. No ratio data are available in the most polluted Milan metropolitan area because only NO$_2$ data are reported. Figure 11b shows the monthly distribution of NO$_x$:NO$_2$ ratio in the Po Valley as grayscale background and the monthly median values as red line. The data coverage is sparse in 2015-2017, and no sensible temporal variation can be identified. Consistent seasonal variation of NO$_x$:NO$_2$ ratio is observable in 2018-2021 with high values (1.5-1.6) in the winter and low values (1.2-1.3) in other seasons, with the caveat that the data after 2020 are not fully validated. The ratios in 2013 and 2014 show a similar seasonal pattern but broader distributions and higher median values in the warm months. Given this discontinuity, we cannot draw a conclusion about inter-annual trend of NO$_x$:NO$_2$ ratio. Nonetheless, the seasonal pattern is robust and consistent with low photochemical reactivity in the winter. Therefore, we average the monthly NO$_x$:NO$_2$ ratios in 2013-2014 and 2018-2019 and use it as a climatology.**

[Figure]

**Figure 11. (a) Black circles are locations of ground-based observation sites where NOₓ and NO₂ data are available. TROPOMI NO₂ TVCD from May 2018 to May 2019 oversampled to 0.02° grid is illustrated in the background. (b) The background shows the density of available NOₓ:NO₂ ratios from filtered hourly ground-based measurements. The red line shows the monthly median values.**

"

Sentences at lines 381-383 of the original manuscript are updated to discuss the limitations of observation-based NOₓ:NO₂ ratio:

"**Monthly climatological NOₓ:NO₂ ratio derived from ground-based observation network is used to convert NO₂ abundance to NOₓ abundance, which improves upon the fixed value used in previous studies (Beirle et al., 2011, Valin et al., 2013, de Foy et al., 2015, Liu et al., 2016). However, uncertainty remains from contamination of NO₂ in-situ measurements (Visser et al., 2019) and the representativeness of surface-based NOₓ:NO₂ ratio to the column-integrated one due to vicinity to emission sources and local ozone titration. Moreover, long-term trend in NOₓ:NO₂ may exist as observed in The Netherlands by Zara et al. (2021), although biases in NOₓ:NO₂ have limited impacts on chemical lifetime and relative emission change estimates.**"

Figure 12 in the original manuscript is updated (now Figure 13 in the updated manuscript) using the monthly NOₓ:NO₂ ratio as shown below. The NOₓ emission rate is spikier in the winter than previous results that assumed a constant NOₓ:NO₂ ratio, but the relative emission changes (panel c) are very consistent. Also note the inclusion of more data in March-June 2021.

[Figure]

*Specific comments*

*L59: it is better to refer to the direct source of information (e.g. Schenkeveld et al. [2017]) on the row anomaly instead of a paper referring to that source.*

Reference added. We keep the reference to Duncan et al. (2016) because we similarly excluded the anomalous rows before the row anomaly happened to keep the spatial sampling consistent.

L63: 'largely' … besides difficulties with a priori profile shapes, there may also be issues with surface albedo and cloud parameters in the TROPOMI NO2 AMF calculation.

Updated to "…, **which can be attributed to the horizontally coarse a priori profile representation as well as uncertainties in surface albedo and cloud parameters in the air mass factor (AMF) calculation.**"

*L112: it is unclear how the uncertainty in the satellite NO2 values has been used. Moreover, it is not recommended to weigh columns according to their uncertainties because the most polluted scenes would then have less influence on the averages. See e.g. discussions on how to calculate representative averages in Miyazaki-papers and Boersma et al. [2016].*

The weight is inversely proportional to the retrieval uncertainty, as used the two references provided (Zhu et al., 2017 and Sun et al., 2018). It is unclear whether the representative error estimates in Miyazaki et al. (2012) and Boersma et al. (2016) are appropriate for this study, because we are not comparing/assimilating satellites to a coarser model grid. Instead, we are oversampling to a much finer grid, and then average within basin boundary. In any case, the dominant error source in the column-wind relationships is the irregular features due to stochastic atmospheric motions and the inadequacy of satellites to sample them, which cannot be improved by changing ways of averaging.

*Figure 4 and 6: can the authors qualitatively explain why April/May has such high NOx emissions and such a low NOx lifetime?*

Concurrence of high emission and low lifetime is problematic. Because the errors of those two are highly anti-correlated, those values are likely unreal and caused by spikes in the column-wind speed relationship in the high-wind speed bins. After regularization is applied, these features are greatly suppressed as in Figure 7. Sentences at lines 187-190 are updated to provide more explanation:

**"Some climatological months (April, May, and September for OMI and August-October for TROPOMI) are characterized by nonphysically high emission rate and low chemical lifetime, whereas others (January and February for OMI) are subject to spurious high chemical lifetime. Those originate from irregular features on the column-wind speed relationship (observable in Figure 3) and tend to be more significant when satellite coverage is low. Because of the stochastic nature of atmospheric motion, those irregular features randomly appear in a limited number of calendar months, leading to wide spread of boostrapping realizations and namely large uncertainties in emission rate and chemical lifetime estimates."**

*L275: indications for shorter lifetimes have also been found over The Netherlands by Zara et al. [2021].*

The following sentence is added after line 278:

**"Over The Netherlands, Zara et al. (2021) found the winter NO$_x$ lifetime decreased from 25 to 19 h and the summer NO$_x$ lifetime decreased from 9 to 8 h using the Chemistry Land-surface Atmosphere Soil Slab (CLASS) model."**

*L305-306: can the authors explain what it is in the instrument performance that causes the reduction in DOFS? Increased noise leading to larger SCD uncertainties? Or would it rather have to do with data availability?*

The OMI data coverage in the Po Valley (figure below) does not show a clear long-term trend, and we consistently exclude row anomaly for the entire OMI record. Hence the gradual drop of DOFS should be from increased noise in level 1 and consequently level 2 data.

[Figure]

This sentence is revised to

**"The decaying DOFS for OMI-based emission rates from 2004 to 2021 is likely due to the gradual increase of OMI radiance noise (Schenkeveld et al., 2017) and consequently increased uncertainties in OMI NO$_2$ TVCD. The higher DOFS from TROPOMI than OMI is also consistent with the instrument performances."**

*L309: please quantify what "match closely" means here.*

This sentence is updated to

**"The long-term trend and seasonality of OMI-based emission rates generally match those from the JPL chemical reanalysis (r = 0.40)."**

*L332: suggest to be careful with words such as "timely". This remains to be seen, the current method may give faster results than bottom-up estimates, but given the heavy level of tuning and need for climatologies to be developed, it cannot be called "fast".*

Please see the first response. In terms of the climatology, it is directly averaged from the observational data and does not consume much time to develop.